# Smart Clothing as a Noninvasive Method to Measure the Physiological Cardiac Parameters

**DOI:** 10.3390/healthcare9101318

**Published:** 2021-10-02

**Authors:** Ching-I Lai, Chang-Franw Lee, Fu-Jin Wei

**Affiliations:** 1Graduate School of Design, National Yunlin University of Science and Technology, Yunlin 640301, Taiwan; 2Department of Fashion Design, Shih Chien University, Taipei 104336, Taiwan; maskerntu@gmail.com; 3Institute of Plant and Microbial Biology, Academia Sinica, Taipei 115201, Taiwan

**Keywords:** smart clothing, comfort, heart rhythm, physiological data device

## Abstract

In response to global aging, there have been improvements in healthcare, exercise therapy, health promotion, and other areas. There is a gradually increasing demand for such equipment for health purposes. The main purpose of smart clothing is to monitor the physical health status of the user and analyze the changes in physiological signals of the heart. Therefore, this study aimed to examine the factors that affect the measurement of the heart’s physiological parameters and the users’ comfort while wearing smart clothing as well as to validate the data obtained from smart clothing. This study examined the subjective feelings of users (aged 20–60 years) regarding smart clothing comfort (within 12 h); the median values were comfortable and above (3.4–4.5). The clothing was combined with elastic conductive fiber and spandex to decrease the relative movement of the fiber that acts as a sensor and increase the user’s comfort. Future studies should focus on the optimization of the data obtained using smart clothing. In addition to its use in medical care and post-reconstructive surgery, smart clothing can be used for home care of older adults and infants.

## 1. Introduction

The MarketsandMarkets Research company analyzed global predictions of wearable devices and projected that the wearable device market has an expected growth of USD 4.72 billion by 2020 at a CAGR of 33.6% between 2015 and 2020 [1]. The global smart textile market is predicted to grow at a rate of 26.2% from 2020 to 2027 [2]. Smart clothing contains conductive fibers or sensors that are attached or woven into the fabric. Similar to the data transfer mechanism used in other wearable devices, data acquired by smart clothing can be transmitted to an APP device, allowing the user to interactively evaluate the data. Since the time smartphones became indispensable in our daily lives, many industries began to intellectualize clothing accessories. This movement has enabled opportunities to introduce wearable technology to our daily living [3]. With advancements in wearable devices, smart clothing has attracted an increasing amount of attention from the market. Apple, Google, and other technology companies have actively been developing smart clothing that integrates electronic components into clothing [2,4]. The developments of new textile materials and miniaturized electronic devices have recently led to the creation of a new generation of smart fabrics. Although these fabrics appear similar to normal fabrics, they can provide specialized functions for various situations as per their design and application. Apart from the addition of electronic components for the intellectualization of fabrics, clothing can be processed to increase its market value with the development of smart textiles [5].

Several types of diverse wearable electronic equipment are available, and the demand for these for the diagnosis of health conditions is also increasing. Compared with traditional clothing, smart clothing provides accurate sensing, display, and transmission functions [6]. Data acquisition is the initial step for smart applications. Human physiological data have been collected, transmitted, and converted with high fidelity using cloud-based platforms in healthcare [7]. Because of successful validation and increased demand, acceptance of smart clothing with medial applications has also increased, thereby expanding the scale of its application [8,9].

Wearable technology is a new frontier in medical monitoring. The application of these wearable devices has gradually entered the clothing industry because of the advances in textile manufacturing techniques and the miniaturization of electronics. In the clothing industry, the development of clothing integrated with various sensors that acquire physiological information is at its early stage but is rapidly evolving. In contrast to other wearable devices, clothing is an item of daily necessity and can be rapidly integrated into consumers’ daily lives. Therefore, smart clothing design should incorporate the accurate acquisition of physiological data, such as electrocardiogram (ECG) data, as well as user comfort to perform long-term monitoring while maintaining a high quality of life for the user [10].

In this study, physiological data acquisition and user comfort while wearing the uniquely designed smart clothing in participants aged 20–60 years were examined. The test method was designed as a framework to validate the utility of the data measured by smart clothing and to understand the factors affecting user comfort. The results of this study will provide invaluable insights into the development of smart clothing for consumers in the healthcare domain.

## 2. Literature Review

### 2.1. Development History of Smart Clothing 

The initial development phase of smart clothing mainly involved combining electronic devices with clothing, with portability and usage being the main objectives. In this phase, popular electronic products were integrated with clothing to endow it with new functions. The 2000 Philips/LevisICD+ series was the world’s first wearable electronic smart clothing [10]; it was also commercially available for consumers. A novel product has been developed by Philips Design, Philips Research, and Levi’s. The 21st-century smart clothing integrates entertainment (such as music) and communication functions [2], with all wires hidden inside the coat. This new series of smart clothing provides a clean and tidy appearance such that users are not hindered by wires while enjoying the convenience of the electronic functions [11].

In smart clothing designed to measure physiological data, wearable sensors are integrated into the textiles, and physiological signals are transmitted through Bluetooth to mobile phones or computers without affecting the user’s comfort. The wearable detection systems used in earlier versions of smart clothing were mainly used to monitor, record, and transmit the physiological parameters of astronauts in space to control stations on Earth. At the 2018 CES Consumer Electronic Fair, smart clothing manufacturer Myant and wireless charger developer Energous collaborated to launch SKIIN innerwear. The innerwear has six sensors integrated into the clothing to detect heart rate (HR), body temperature (BT), psychological stress, movement, body fat percentage, and water content. These physiological parameters are transmitted to a mobile phone. A unique characteristic of this clothing is that it uses wireless charging [2].

The future of smart clothing may include applications ranging from entertainment and personalized fitness to value-added healthcare. However, a major challenge is to meet consumer expectations for smart clothing, which include convenience, comfort, and softness while providing integrated high-technology function [12]. Therefore, this study examined the feasibility of smart clothing in healthcare monitoring functions as well as the user comfort of smart clothing in daily living.

### 2.2. Conductive Silver Fibers

Conductive silver fibers are specialized products produced by permanently binding a layer of pure silver on fiber surfaces. This layered structure not only enables the fiber to retain its original textile characteristics but also endows it with electrical conductivity due to the silver coating [13]. Silver fiber synthesis uses nanosilver wires [14,15,16] as the electrically conductive carrier. Sputtering is used to enable the clothes to conduct electricity. This process can be used to produce or control electricity, light, heat, and other physical qualities. Silver ion fibers also possess antibacterial activity, which can help avoid secondary infections. The compound spinning method is used to balance relaxation and adhesion forces between the silver fibers and polymers, allowing effective antibacterial activity. This process is applied to highly elastic conductive fibers that are lightweight and comfortable enough to allow free body movement. The smart clothing conductive fabric is a composite fiber that must conform to ANSI/AAMI EC-12 regulations (i.e., impedance <2 kiloohms) [13]. The most commonly used synthetic fibers are polyester and nylon; as these are soft and elastic, knitting is an ideal method for the development of wearable clothing. Single jersey and interlock knitted fabrics as well as blended fabrics, including silver fiber, polyester, spandex, and nylon, are used. The elastic fibers in these materials can increase the stability of sensors when worn on the body.

Smart clothing is designed to be comfortable, breathable, and moisture-permeable, and it can be worn with other clothes. The tight-fitting nature of this clothing is used to fix the electrodes in place, and the use of silver fiber fabrics as ECG electrodes decreases the chances of skin damage [17,18]. Smart clothing ensures that the user is not affected by monitoring during data acquisition, thus making it suitable for long-term cardiac monitoring. Smart clothing is a suitable alternative to traditional disposable ECG electrodes.

Therefore, blending different ratios of silver fibers can translate into considerations for data signal, costs, and wearability. Furthermore, silver fiber possesses antibacterial activity that can prevent secondary infection, which can be critical in healthcare and long-term care.

### 2.3. Comfort in Smart Clothing 

Comfort is a neutral characteristic of clothing, which means that wearing clothes can protect the user without any physical and mental effects, i.e., it can rapidly promote a sensation of warmth. Comfort is a state of not feeling excessively cold or warm. In other words, there should be a supply–demand balance in terms of the heat generated and lost by the body to maintain the ideal temperature. User comfort specifically includes the following domains: (1) thermal comfort, (2) contact comfort, (3) pressure comfort, and (4) clothing aesthetics. In the human body–clothing–environment system [19], humans are homoiotherms and will employ various means to dissipate heat and moisture to maintain a stable temperature. Therefore, clothes play a critical role in temperature regulation.

The comfort of the fabric can be divided into two components. The first is physiological comfort, which affects the surface temperature, amount of sweat and heat dissipation, and other physiological responses associated with heat generation and dissipation. The second component is psychological comfort, i.e., recollection of past or preconceived experiences; this component is a subjective feeling that is assessed and compared using questionnaire surveys. Subjective evaluations are based on the use of the senses (vision, touch, thermesthesia, and humid heat sensation) to evaluate and describe physiological and psychological comfort [20,21]. Therefore, verbal responses are used to understand subjective feelings; this method can effectively reflect the actual feelings of the users. In general, clothing comfort can be tested in a temperature- and humidity-controlled room. Different levels of activity and changes in the testing environment can be used to control these factors. Clothing parameters based on questionnaires can be effectively used to measure contact comfort and understand subjective factors, such as material, touch, tight-fitting lines, hygroscopicity, breathability, and mobility. In addition to contact comfort, the users’ feelings regarding the clothes can also be used to assess the comfort in varying humidities and temperatures.

### 2.4. Physiological Data Device

This study used smart clothing to measure physiological signals using snap-on physiological signal transmission devices. The smart clothing, signal control unit, and signal sensing unit are three major components of this system (Figure 1). The signal control unit includes a module for data transmission, processing, and calculation. ECG signals are acquired by the smart clothing and undergo low-pass wave filtration through the MAX30001 module along with signal amplification before being turned into digital ECG signal output. MAX30001 can generate ECG waveform, HR, and rhythm. This device also contains a BioZ channel that measures respiration. Biopotentials and BioZ channels provide high input impedance values, low noise, a high common mode rejection ratio, programmable gain, multiple low-pass and high-pass wave filters, and a high-resolution analog-to-digital converter. The two channels contain electrostatic discharge, protection, electromagnetic interference, filters, inner lead biasing, DC disconnection detection, connection detection in standby mode, and many calibration voltages for self-testing. The start sequence protocol ensures that unwanted transient events are not detected by the electrode [22].

ECG is a medical test commonly used to screen patients’ cardiac function by analyzing cardiac electrical activities. P, Q, R, S, and T waveforms, known as the PQRST complex, are detected by the ECG, and the PQRST complex is the waveform of one heartbeat cycle (Figure 2).

HR is calculated on the basis of the R–R interval between the QRS complexes; the formula to calculate HR from the R–R interval is as follows (Figure 2):HR = 60/R–R interval (in sec),
where the HR unit is bpm.

MAX3001 is equipped with software that calculates the real-time cardiac parameter, which includes analysis of the HR and PQRST waveforms.

### 2.5. Heart Rhythm

In general, there are four commonly monitored vital signs: BT, blood pressure (BP), heart rhythm (HRm), and respiratory rate (RR) [23]. These parameters are used to evaluate the patient’s physical well-being by providing evidence for the presence of latent disease and recovery progress. The normal vital signs of each individual vary with age, weight, sex, and overall health [24]. ECG tracing of a heartbeat typically includes a P wave, a QRS complex, and a T wave. An ECG machine using gel electrodes can be used to measure myocardial potentials, which are converted into an ECG waveform showing P, Q, R, S, and T waves [25]. Analysis of HRm includes HRm variability (HRV), which refers to physiological variance between two consecutive R–R intervals or the distance between two consecutive R waves [26]. Frequency spectrum analysis is a reliable HRV evaluation method that is influenced by the autonomic nervous system. HRV analysis can reflect the condition of the cardiovascular system [4]. For example, HR increases with an increase in exercise intensity and gradually decreases after exercise cessation.

In the analysis of HRm, one must consider several causes of ECG signal interference that are associated with the user, electrodes, environment, and device. These factors interfere or negatively impact impedance in the areas around the electrode’s point of contact with the skin [27,28]. Traditional methods of measurement involve gel electrodes establishing effective contact between the electrodes. Electrodes have often interfered with the patient’s sleep quality and, ultimately, health outcomes [28].

This study examined vital signs, including BT, BP, HRm, and RR. The physiological cardiac parameters obtained using smart clothing and those obtained using conventional gel electrodes were compared in this study.

## 3. Materials and Methods

### 3.1. Data Collection

#### 3.1.1. Study Design

This study aimed to determine whether the physiological cardiac parameters obtained from smart clothing and 3M gel electrode patches were comparable. The inclusion criteria were Taiwanese residence and an age of 20–60 years. Five males and five females participated in the study. The study content and objectives were explained to the subjects before the experiment, and their informed consent was obtained. None of the participants had any significant medical conditions. Data acquisition was performed in three phases.

Phase 1: Three materials were used. A: This was made of 100% silver fiber using a single jersey knitting method. It had the highest ratio of silver fiber to increase the skin contact area compared with the other materials. B: The material was made of 75% silver fiber, 23% polyester, and 2% spandex using a single jersey knitting method. Elastic spandex fibers were added to increase tightness. C: The material was made of 50% silver fiber and 50% nylon using the French terry knitting method to decrease material costs. (Shown below as A, B, and C; Figure 3). The electrical conductivity of the three materials was evaluated, and the most suitable conductive material was used to design the smart clothing.

Phase 2: Smart clothing fitted for each subject was worn for 12 h at home. A 5-point Likert scale (i.e., extremely comfortable, comfortable, fair, uncomfortable, and extremely uncomfortable) was used to assess their responses in terms of material, tactile sensation, tight-fitting lines, hygroscopicity, breathability, and mobility.

Phase 3: MAX30001 was used to measure the physiological cardiac parameters of the smart clothing. Simultaneously, 3M gel electrode patches were used to measure the same parameters. A total of 23,040 datasets were generated by the smart clothing and 3M gel electrode patches during 3 min interval measurements. The HR datasets from the smart clothing were compared with the datasets from the 3M gel electrode patches.

#### 3.1.2. Smart Clothing Design

The smart clothing used fabric that combines conductive silver fibers and traditional highly elastic spandex fibers. This type of hybrid fabric minimizes the relative movement between the conductive silver fibers and the skin to optimize the signal-to-noise ratio of the data and to increases user comfort. Three metallic button electrodes were fixed in the front chest and waist in the conductive cloth to sense and acquire the data. A zipper was added in front of the smart clothing to increase the ease of wearing and removing the clothing (as shown in Figure 4). Three electrodes, labeled as BIN, BB, and BP, were designed as part of the smart cloth to transmit data, measure signals, and calculate potential changes for conversion to real-time data, respectively.

### 3.2. Data Analysis

This study was conducted indoors where the temperature and relative humidity were maintained at 26 °C–28 °C and 55–65%, respectively. Subjects were required to rest for 10–15 min after entering the test environment. They were asked to lay in a supine position in a relaxed manner (Figure 5). Before beginning data acquisition, the aim and method of the test were explained to each subject. As baseline data, all subjects completed 3 min monitoring of the physiological cardiac parameters using smart clothing and 3M gel electrode patches. After this, 23,040 (128 sets × 180 s) HRm datasets were generated as test samples. After the baseline test was completed, the subjects responded to the questionnaire on the comfort of the smart clothing.

## 4. Study Results

### 4.1. Analysis of the Conductive Materials

#### 4.1.1. Confirmation of Test Environment and Device Stability

The stability of the test device can be affected by ambient temperature and humidity. Therefore, we performed a device under test (DUT) at different time points and environmental conditions to confirm its stability. The DUT was performed using an AWG28 nickel wire. The temperature coefficient ratio (TCR), length, test numbers, test interval, and device used were 15 cm, 1000, 10 s, and Keysight 3458A Digital Multimeter, respectively. The results are shown in Table 1. In the same DUT, the TCR of the AWG28 wire [16] on different days was unchanged at 0.0059 ohms [29]. When the temperature was increased by 2 °C (test 2), the actual TCR value was 0.0433, in contrast to the theoretical or calculated value of 0.0429 ohms; this was only a difference of 0.0004 ohms (Table 1). Consequently, the Keysight 3458A Digital Multimeter was found to be a stable test device.

The impedance formula for the conductor is as follows:R = Rref [1 + α (T − Tref)],
where 

R = conductor resistance at temperature (T),

Rref = conductor resistance at reference temperature,

α = temperature coefficient of resistance for the conductor material,

T = conductor temperature (°C), and

Tref = reference temperature at which α is specified for the conductor material. Its value is usually 20 °C, but may sometimes be 0 °C.

#### 4.1.2. Analysis of the Three Conductive Materials

To examine the accuracy of the data measured by the smart clothing, we selected three types of conductive material to measure the impedance or ohm values. The three conductive cloths (A, B, and C) were made by blending different materials and variables, including ambient temperature, humidity, and distance, that could theoretically affect impedance. Different ratios of silver fibers were used in these three materials. These differences translated into greater impedances of materials B and C compared with A. The Keysight 3458A Digital Multimeter was used to obtain the corresponding mean ohms, temperature, and humidity of the three materials. These materials were tested at 1 cm, 2 cm, 5 cm, and 10 cm. Table 2 shows the test results. We chose material B because of the reduced cost of blended weaving, easy mass production, and within-range conductivity.

### 4.2. Data Analysis Methods

Demographics and descriptive statistics of smart clothing comfort: First, demographics of the subjects’ backgrounds and qualitative evaluation-related data were conducted. Using analysis of variance (ANOVA), the independent variables were gender, age, and education level; the subjective response to wearing the smart clothing was the dependent variable. Microsoft Excel, R, and SPSS statistical software were used for data analysis. Descriptive statistics and ANOVA were also used for statistical analysis in this study. For adjusting the small number of subjects, we used the traditional (Appendix A) and Bayesian ANOVA (Appendix A) as well as the Student’s (Appendix A) and Bayesian *t*-tests [30] (Appendix A). The significance level was set at 0.05. All the study data conformed to the normality assumption.

#### Smart Clothing Comfort Analysis

The feeling of the material, tactile sensation, tight-fitting lines, hygroscopicity, breathability, and mobility were scored from 1 to 5 points based on a Likert scale. The statistical analysis revealed that there were no significant differences in subjective feelings toward smart clothing comfort (12 h), smart clothing material, tactile sensation, tight-fitting lines, hygroscopicity, breathability, and mobility (*p* > 0.05). The mean scores of these items were in the fair to extremely comfortable (3.4–4.5) range (Table 3). The median values were comfortable and above, except the median breathability score in the appropriate category.

### 4.3. Analysis of the Physiological Cardiac Parameters

Data analysis of the 10 subjects was sampled for physiological cardiac parameters. The plot function in the Max30001 evaluation system was used to measure the number of 3 min HR intervals as 23,040 datasets (Figure 6). The data obtained after averaging is presented in Table 4. The paired-sample *t*-test was used to determine whether there were significant differences between smart clothing and gel electrodes (Table 5). The Bayesian paired sample *t*-test results of the data obtained from smart clothing and gel electrodes found no significant difference in the HR between the two methods (Bayes factor: 0.3089). The Bayes factor (<1) indicated acceptance of the null hypothesis, thus showing no significant differences.

## 5. Discussion

In the current healthcare model, the gel ECG electrode patch is the standard of care to noninvasively obtain physiological cardiac parameters, i.e., HR and HRm. Unfortunately, skin irritation, the high cost of nonreusable ECG patches, and the long duration of patch placement can be limiting factors. This study focused on the evaluation of a novel noninvasive alternative method to measure the physiological cardiac parameters using smart clothing. A paired-sample *t*-test was used to determine the differences in physiological cardiac parameters measured using smart clothing and 3M gel ECG electrode patches (Appendix A). To assess the functionally of the smart clothing, we also examined the subjective factors affecting smart clothing comfort using a 5-point Likert scale and a questionnaire survey.

Yoder et al. [31] analyzed 54,096 patients and found that emergencies did not occur within 24 h in 45% of patients who were awakened at night to measure their vital signs. If patients’ sleep is not interrupted to change ECG electrodes, their quality of life could be significantly better because of uninterrupted sleep, which translates into better emotional states, speedier recovery, and shorter hospitalization durations.

Smart clothing offers the advantages of monitoring vital signs at night without interrupting sleep while maintaining safety, which can facilitate long-term monitoring of physiological cardiac parameters in older adults living alone. In addition to the excellent quality of the ECG tracings and fidelity of the HR compared with the traditional ECG patches, this study also demonstrated the subjects’ comfort, which translates to long-term wearability and compliance.

Of the three silver fiber conductive fabrics, the mean ohm value of conductive fabric A was 0.1983, which was the best value, but it also had the highest production cost among the three fabrics. Fabric C had the lowest silver fiber ratio and the weakest mean ohm value (2.2395), whereas the mean ohm value of Fabric B was 0.8806, which appeared to be the most optimal. Elastic spandex fibers were added to Fabric B to increase comfort and decrease material costs; it was considered the best prototype for future commercialization.

Once the fabric material was determined, we considered the comfort of the smart clothing design. A smart clothing zipper was designed for easy donning and doffing. Three metal button electrodes were embedded in the silver fiber to substitute for the gel electrodes.

This initial study primarily focused on the measurements of the physiological cardiac parameters as a proof of concept of the smart clothing. Therefore, there are two notable limitations to this study: the small sample size and the education level of the subjects. In addition to studying the 10 subjects, we performed the tests on 10 residents who were relatively more well-educated than the general population. Higher education level translated into greater ability to absorb instructions for the appropriate use of smart clothing. By contrast, the attitudes and understanding of subjects with lower education levels were not evaluated. Moreover, our subjects were primarily in the age group of >60 years, which can potentially translate to variability in the data due to physiology, body type, and/or aesthetic perception of the clothing.

## 6. Conclusions

In this study, the mean values for subjects wearing the smart clothing were comfortable and above on the Likert scale. However, the number of subjects who wore smart clothing in our study was insufficient for generalization. There were no quantitative or qualitative differences in the HR and ECG tracings between those measured using smart clothing and traditional 3M gel electrode patches. On the basis of the results of this study, we believe that additional physiological parameters, such as BT, BP, and RR, can be incorporated into future studies. It is our conclusion that smart clothing can potentially have long-term and practical healthcare applications in inpatient and outpatient settings.

## Figures and Tables

**Figure 1 healthcare-09-01318-f001:**
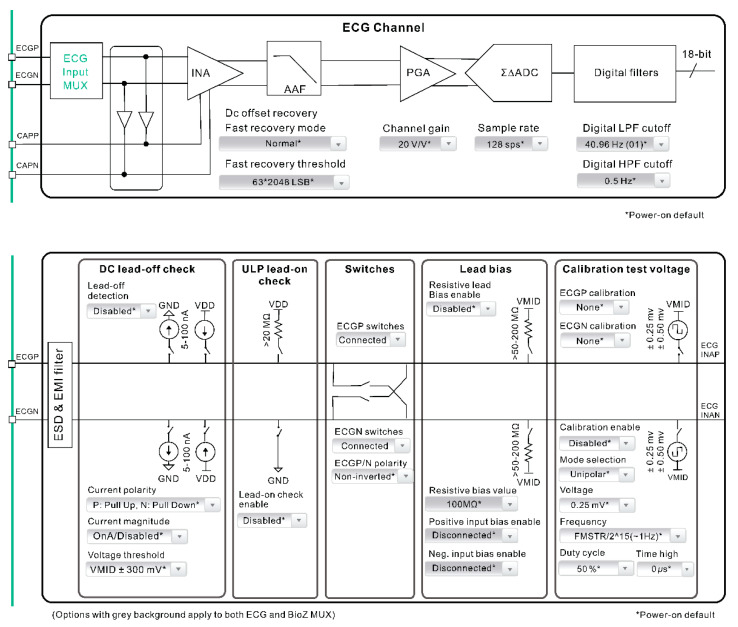
Schematic diagram of smart clothing data measurement framework.

**Figure 2 healthcare-09-01318-f002:**
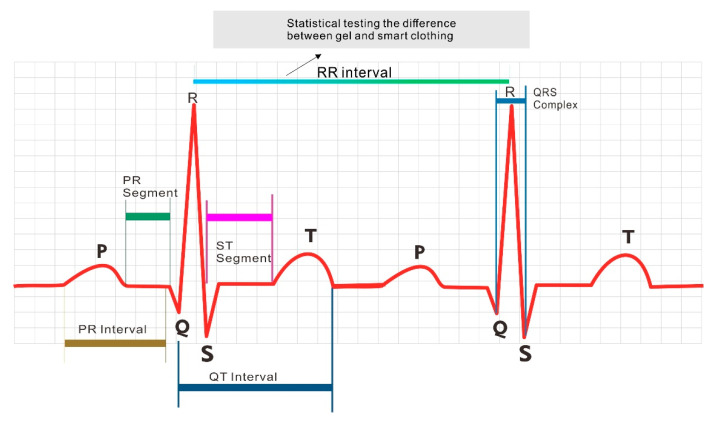
Heartbeat cycle waveform and R–R interval.

**Figure 3 healthcare-09-01318-f003:**
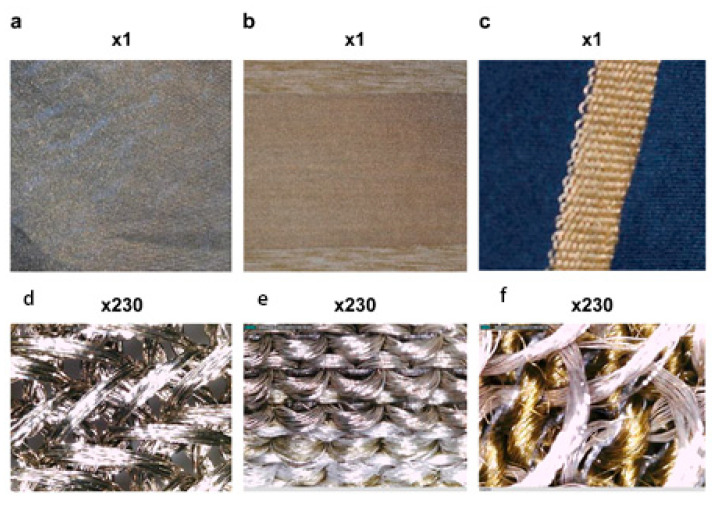
The three types (**a**–**f**) of conductive material (1× and 230× magnification).

**Figure 4 healthcare-09-01318-f004:**
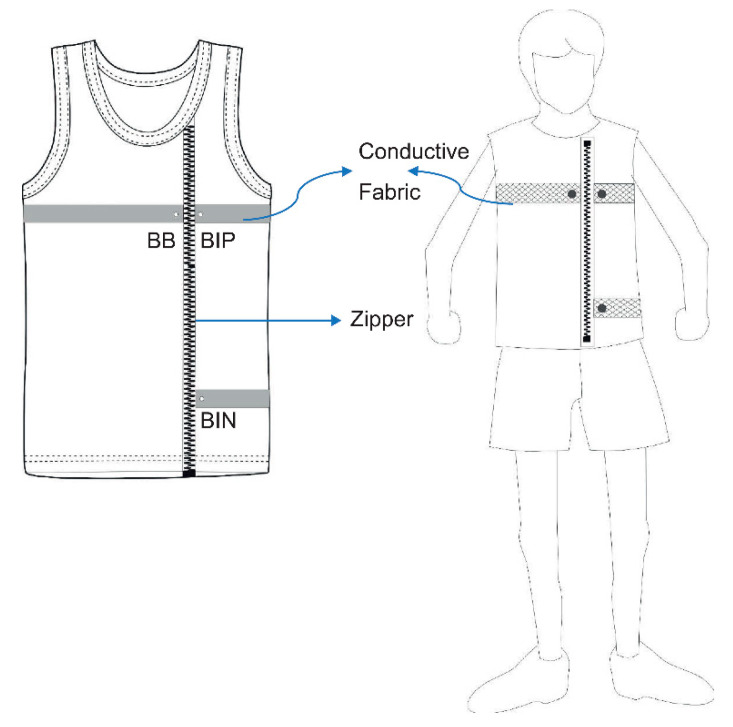
Smart clothing design.

**Figure 5 healthcare-09-01318-f005:**
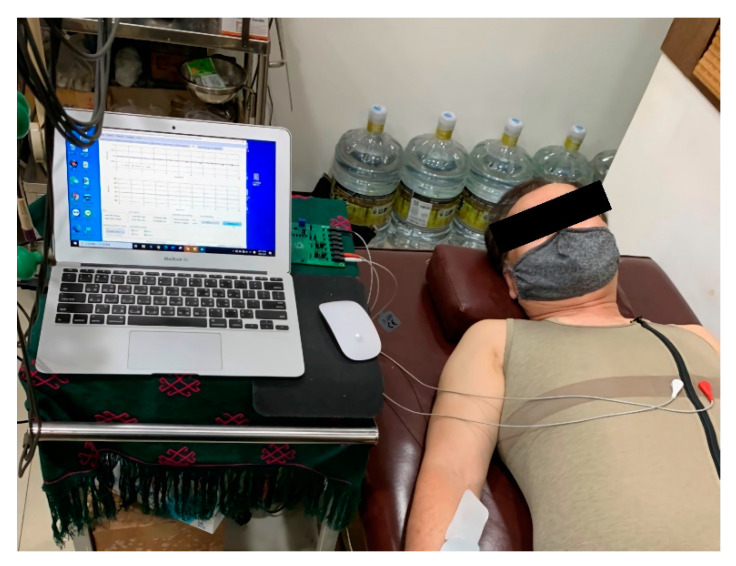
Subject rests in a supine position on a bed in a relaxed manner.

**Figure 6 healthcare-09-01318-f006:**
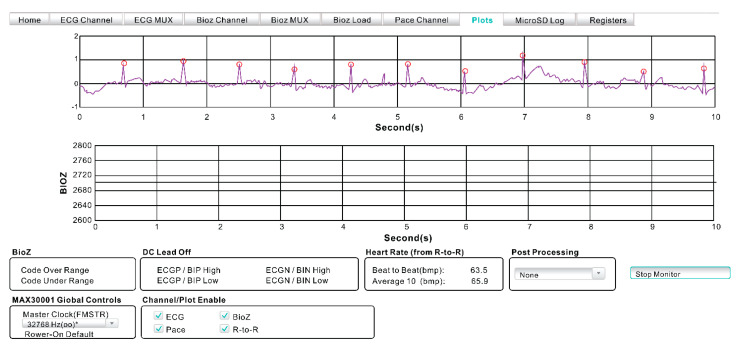
Raw ECG tracing from the MAX30001 device.

**Table 1 healthcare-09-01318-t001:** Corresponding temperature, humidity, and mean ohm values of the two tests.

Test	Temperature (°C)	Relative Humidity (%)	Mean Ohm
Test 1	26.0–27.1	65.0–71.3	0.0419
Test 2	28.4–29.7	56.9–59.3	0.0433

Data source: Compiled from this plan.

**Table 2 healthcare-09-01318-t002:** Corresponding temperature, humidity, and mean ohms of the three conductive materials.

Material	Temperature (°C)	Relative Humidity (%)	Mean Ohms (2 cm)
A	27.95–28.06	58.0–60.1	0.1983
B	26.70–27.09	61.2–62.3	0.8806
C	26.50–26.80	61.6–62.6	2.2395

**Table 3 healthcare-09-01318-t003:** Q1–Q6 means and standard deviations.

	Question	Minimum	Maximum	Mean	Standard Deviation
Q1. What is the degree of subjective feeling that the “material” of the smart clothing gives you?	10	3.0	5.0	4.3	0.67
Q2. What is the degree of subjective feeling that the “tactile sensation” of the smart clothing gives you?	10	4.0	5.0	4.5	0.53
Q3. What is the degree of subjective feeling that the “tight-fitting lines” of the smart clothing gives you?	10	2.0	5.0	3.7	0.82
Q4. How do you feel about the “hygroscopicity” of the smart clothing when worn?	10	2.0	4.0	3.6	0.70
Q5. How do you feel about the “breathability” of the smart clothing when worn?	10	2.0	5.0	3.4	0.84
Q6. How do you feel about the “mobility” of the smart clothing when worn?	10	3.0	5.0	4.5	0.85

**Table 4 healthcare-09-01318-t004:** Mean 3 min interval of the physiological cardiac parameters.

S/N	Age	Smart Clothing Beat to Beat (bpm)	Gel Beat to Beat (bpm)
M01	82	68	69.2
M02	68	68	68.6
M03	70	66.8	65.8
M04	50	68	65.8
M05	40	64.5	64.5
F01	52	66.8	68.6
F02	55	73.8	74.6
F03	66	70.5	69.2
F04	64	70.5	70.5
F05	45	68	68

**Table 5 healthcare-09-01318-t005:** Paired-sample testing of smart clothing and gel electrodes.

	Bayesian Paired *t*-Test ^※^					t	df	Bayes Factor
	Mean	Standard Deviation	Standard Error of the Mean	95% Confidence Interval of the Difference				
				Lower Limit	Upper Limit			
Paired Cloth_bpm—Gel_bpm	−0.0020	0.3569	0.0113	−0.5344	0.5533	1.6948	9	0.3089 ^†^

**^※^** Bayesian *t*-test performed using Bayes Factor [30], a package in R. Iteration time = 1000. ^†^ The Bayes factor (<1) indicated acceptance of the null hypothesis, thus showing no significant differences.

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
