# Peer review of "Smart Clothing as a Noninvasive Method to Measure the Physiological Cardiac Parameters"

_healthcare, 2021, doi:10.3390/healthcare9101318_

Round 1

Reviewer 1 Report

This study, which describes the use of new connected clothing for health monitoring, is very interesting. It is relatively well written and clear. 
We can regret that the number of subjects remains low, it significantly decreases the value of the proposed results. 
The statistical tests remain relatively simple and the exploitation of the cardiac data could have been more detailed by analyzing for example the RR signal and its different indices. Indeed, one can wonder if the analyses in time, frequential and non linear domains of cardiac indices could also be exploited from the biosignals collected with these clothes. This would bring an interesting added value to the study and would allow to go further in the monitoring of the health or well-being of people. Recently, recently authors propose to study non-linear indices (fractality, entropy) in order to evaluate the psychophysiological state of people. Can this be envisaged from the biosignals collected? In this perspective, some bibliographical references should be added...

Author Response

Reviewer  1 Comment:
This study, which describes the use of new connected clothing for health monitoring, is very interesting. It is relatively well written and clear. 
We can regret that the number of subjects remains low, it significantly decreases the value of the proposed results. 
The statistical tests remain relatively simple and the exploitation of the cardiac data could have been more detailed by analyzing for example the RR signal and its different indices. Indeed, one can wonder if the analyses in time, frequential and non linear domains of cardiac indices could also be exploited from the biosignals collected with these clothes. This would bring an interesting added value to the study and would allow to go further in the monitoring of the health or well-being of people. Recently, recently authors propose to study non-linear indices (fractality, entropy) in order to evaluate the psychophysiological state of people. Can this be envisaged from the biosignals collected? In this perspective, some bibliographical references should be added...

Response: thank you to our reviewers for your comments and opinions, we were encouraged to increase the number of our research subjects to 45 people and a biological signal using the analysis time, and nonlinear frequency-domain index of heart Because it takes more time to collect data, we expect to complete it in 60-90 days. Therefore, after discussion, we have revised our statistical analysis method for accuracy.
The psychological research part is limited in space and we will implement it in the next stage
As for the English translation, we have asked a professional translation agency to revise the text to make the article language more fluent.

Reviewer 2 Report

The articel 'Use of Smart Clothing as a Noninvasive Method to Measure the Physiological Parameters of the Heart' has an interesting topic. The interesting topic is an area that can be pay more effort to do research in. There is a excellent literature review to intruduce the back ground of this article. However, there are, in my opinion, several problems in this paper. 

  1. There are not enough scientific soundness in this article. Although this area is an interesting area, the author does not describe enough scientific methods to analyse the data.
  2. There are not enough results of experiments to support the opinion of the author.
  3. This article is about analysing the physiological characteristic of smart clothing, but there is little demonstration of this part.

Author Response

Reviewer 2 Comment:
The articel 'Use of Smart Clothing as a Noninvasive Method to Measure the Physiological Parameters of the Heart' has an interesting topic. The interesting topic is an area that can be pay more effort to do research in. There is a excellent literature review to intruduce the back ground of this article. However, there are, in my opinion, several problems in this paper. 
1.    There are not enough scientific soundness in this article. Although this area is an interesting area, the author does not describe enough scientific methods to analyse the data.
2.    There are not enough results of experiments to support the opinion of the author.
3.    This article is about analysing the physiological characteristic of smart clothing, but there is little demonstration of this part.

Response: We have added more demonstration photos and revised the analysis process for collecting physiological data of smart clothes. We used the traditional ANOVA and Bayesian ANOVA, as well as the Student t-test and Bayesian t-test
As for the English translation, we will ask a professional translation agency to revise the text to make the article language more fluent.
Thank you again for your comments 

Round 2

Reviewer 1 Report

Thank you for taking into account my remarks, especially about the statistical method. Despite the relatively small number of subjects, I think this study is interesting enough to be published as is.

Author Response

Reviewer 1 Comment:

Comments and Suggestions for Authors

Thank you for taking into account my remarks, especially about the statistical method. Despite the relatively small number of subjects, I think this study is interesting enough to be published as is.

Response: thank you for accepting our research.

Reviewer 2 Report

  • The grammar and vocabulary of the paper are not standardized and need to be polishedï¼›
  • Figure 2 and Figure 3 are proposed to be merged.ï¼›
  • All figures in the papei are recommended to be drawn as vector diagrams. 

Author Response

Reviewer 2 Comment:

The grammar and vocabulary of the paper are not standardized and need to be polishedï¼›

Figure 2 and Figure 3 are proposed to be merged.ï¼›

All figures in the papei are recommended to be drawn as vector diagrams.

Response: Thank you for your valuable comments. We have revised the figures as advised.

We have also got our manuscript edited again to ensure there are no errors in the grammar and vocabulary.
